# RE-DENSIFICATION MEETS CROSS-SCALE PROPAGATION: REAL-TIME NEURAL COMPRESSION OF LIDAR POINT CLOUDS

## ABSTRACT

LiDAR point clouds are fundamental to various applications, yet high-precision scans incur substantial storage and transmission overhead. Existing methods typically convert unordered points into hierarchical octree or voxel structures for dense-to-sparse predictive coding. However, the extreme sparsity of geometric details hinders efficient context modeling, thereby limiting their compression performance and speed. To address this challenge, we propose to generate compact features for efficient predictive coding. Our framework comprises two lightweight modules. First, the Geometry Re-Densification Module re-densifies encoded sparse geometry, extracts features at denser scale, and then re-sparsifies the features for predictive coding. This module avoids costly computation on highly sparse details while maintaining a lightweight prediction head. Second, the Cross-scale Feature Propagation Module leverages occupancy cues from multiple resolution levels to guide hierarchical feature propagation. This design facilitates information sharing across scales, thereby reducing redundant feature extraction and providing enriched features for the Geometry Re-Densification Module. By integrating these two modules, our method yields a compact feature representation that provides efficient context modeling and accelerates the coding process. Experiments on the KITTI dataset demonstrate state-of-the-art compression ratios and real-time performance, achieving 26 FPS for encoding/decoding at 12-bit quantization. The code will be publicly available upon acceptance.

## 1 INTRODUCTION

With the rapid advancement of 3D sensing technologies, massive amounts of point cloud data have been accumulated in various fields such as autonomous driving and mapping You et al. (2020). This surge in data volume has led to an increasing demand for precise point cloud compression (PCC). Currently, most PCC methods represent raw coordinate data using quantized structures such as range images (Wang et al., 2022; Zhou et al., 2022; Wang & Liu, 2022; Stathoulopoulos et al., 2024), voxels (Quach et al., 2019; He et al., 2022; Wang et al., 2025; Yu et al., 2025), or octrees (Biswas et al., 2020; Huang et al., 2020; Que et al., 2021; Chen et al., 2022; Fu et al., 2022; Song et al., 2023), and then apply techniques like prediction or transformation to achieve compression.

Although existing PCC methods have made significant progress in rate-distortion (RD) performance, their foundational representations, voxels or octrees, exhibit inherent limitations in high-precision compression scenarios. Both representations quantize a 3D space into discrete volumes, marking each as occupied only if it contains at least one point. However, as shown in Fig. 1a and Fig. 1b, with the quantization resolution increases, the local neighborhood around a given voxel becomes increasingly sparse, drastically reducing the availability of contextual information. We term this phenomenon as **High-Resolution Contextual Sparsity (HRCS)**. In such cases, predicting the occupancy of a given voxel becomes particularly challenging due to the sparsity of context. However, simply enlarging the receptive field typically incurs substantial computational overhead (where the receptive field will grow cubic in 3D space), making it impractical for efficient compression.

To quantify HRCS, we conducted data statistics on all frames of the KITTI dataset. For the octree of each sample, we collected two key statistics: (i) the total number of nodes at each level, and (ii) the

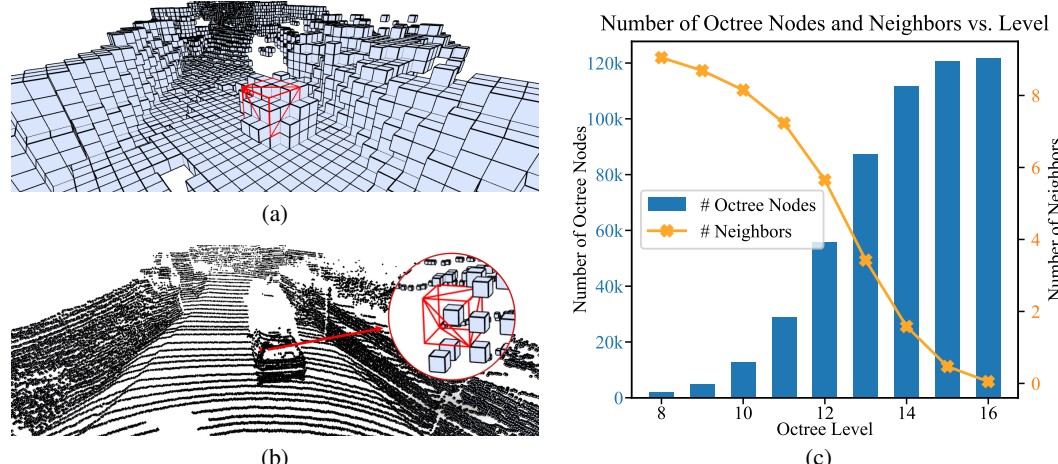

Figure 1: Illustration of High-Resolution Contextual Sparsity (HRCS) phenomenon: (a) and (b) depict the voxelized octree representations at levels 8 and 12 for a point cloud from the KITTI dataset, respectively. The red bounding box highlights a $3 \times 3 \times 3$ neighborhood centered at the same spatial location. As the resolution increases, the number of valid context nodes within this neighborhood drops sharply, from 21 nodes at level 8 to zero at level 12, which illustrates the emergence of HRCS. (c) quantifies HRCS on the KITTI dataset, where the average number of neighbors per node decreases sharply with increasing octree level.

average number of occupied neighbors within a $3 \times 3 \times 3$ cube centered at each node. As illustrated in Fig. 1c, with increasing resolution (*i.e.*, at deeper octree levels), the growth rate of the number of nodes slows down significantly, while the average number of neighbors per node drops sharply. At certain levels, the average number of neighbors even falls below one. Moreover, this decline exhibits a marked inflection point at a specific octree level, indicating a nonlinear loss of contextual richness.

To solve the HRCS problem without compromising coding efficiency, this paper proposes a Geometry Re-Densification (GRED) strategy. Specifically, given the task of encoding nodes $\mathbf{X}^l$ at level $l$, GRED first traces back to a shallower level $k$, where $\mathbf{X}^k$ retains relatively denser neighborhood features. $\mathbf{X}^k$ is then reverted to the original sparse domain through a series of lightweight convolutions and upsampling operations. These features are spatially aligned with $\mathbf{X}^l$ which are subsequently utilized to facilitate the occupancy prediction of $\mathbf{X}^l$. Upon GRED, this paper proposes a Cross-Scale Feature Propagation (XFP) module to better leverage information across different resolution levels. Specifically, XFP combines dense features from shallow levels with sparse features from deeper levels. The sparse features are first densified using GRED. Then, the features from both levels are fused to predict the occupancy of octree nodes. This cross-scale fusion enables more accurate predictions under sparse contextual conditions while maintaining computational efficiency.

In the following sections, we first review related work in PCC, followed by a detailed description of GRED and XFP in the proposed method. In the experimental section, we evaluate our method on two widely used datasets, KITTI (Geiger et al., 2012) and Ford (Pandey et al., 2011). The results demonstrate the effectiveness and superiority of the proposed method.

## 2 RELATED WORK

This section summarizes the representative point cloud compression works up to now. According to the different representation methods of point cloud data during the compression process, we classify most of the existing PCC schemes into two categories: 1. Voxel-based PCC; 2. Octree-based PCC.

**Voxel-based PCC**. Voxel-based approaches split the point cloud into sufficiently small voxels, utilizing sparse convolution (Tang et al., 2023) to optimize memory usage. Based on the voxel, many PCC techniques have emerged (Wiesmann et al., 2021; Nguyen et al., 2021; Tzamarias et al., 2022; Nguyen & Kaup, 2022; Pang et al., 2024; Zhang et al., 2025a; Meng et al., 2025; Zhang et al., 2025b). For example, Wang et al. (2021) proposed a voxel-based geometry compression method

that partitions point clouds into non-overlapping 3D cubes and leverages a variational autoencoder-driven convolutional neural network to extract latent features and hyperpriors for entropy coding. Recently, Wang et al. (2025) proposed a universal multiscale conditional coding framework, Unicorn, which leverages sparse tensors from voxelized point clouds and cross-scale temporal priors to enhance geometry compression. Zhang & Gao (2025) proposed a dynamic point cloud compression framework based on voxelized data, featuring a slimmable architecture with multiple coding routes for rate-distortion optimization, and a coarse-to-fine motion module to improve inter-frame prediction.

**Octree-based PCC**. Octree-based approaches typically construct an $L$ level octree by recursively subdividing the point cloud within a pre-defined bounding volume, and achieve compression by predicting the occupancy status of each octree node. Based on the octree structure, many PCC techniques have emerged (Kammerl et al., 2012; Golla & Klein, 2015; Garcia & de Queiroz, 2017; Wen et al., 2020; Luo et al., 2024). For example, Huang et al. (2020) proposed an octree-based compression method that leverages a tree-structured conditional entropy model to exploit sparsity and structural redundancy in LiDAR point clouds. Similarly, Fu et al. (2022) proposed an octree-based deep learning framework that encodes point clouds by leveraging rich sibling and ancestor contexts with an attention mechanism. Cui et al. (2023) proposed OctFormer, which constructs node sequences with non-overlapping context windows and shares attention results to reduce computation. Song et al. (2023) proposed an octree-based entropy model with a hierarchical attention mechanism and grouped context structure, reducing the complexity and decoding latency of large-scale auto-regressive models.

**In summary**, both voxel-based and octree-based PCC approaches have seen substantial progress, with learning-based methods outperforming traditional handcrafted-feature approaches (Mekuria et al., 2017; Schwarz et al., 2019; Garcia et al., 2020; Song et al., 2021; Wang et al., 2022; Qin et al., 2024; Cao et al., 2025) in terms of rate-distortion performance. However, under high-resolution settings, both representations tend to suffer from HRCS. This sparsity significantly limits the effectiveness of feature learning and representation. Despite its impact, this challenge remains largely underexplored in current research.

## 3 METHOD

To address the HRCS problem and meet the requirements of real-time LiDAR PCC, this paper proposes a fast encoding framework based on octree representation. The overall architecture is illustrated in Fig. 2. The proposed framework comprises four key components: octree construction, prior construction, cross-scale feature propagation, and entropy coding. In particular, this section provides a detailed introduction to the Cross-Scale Feature Propagation module, with an emphasis on its core component, namely the Geometry Re-Densification module.

### 3.1 GEOMETRY RE-DENSIFICATION MODULE

The irregular and unordered nature of LiDAR point clouds poses significant challenges for efficient processing on modern hardware architectures. To better exploit existing hardware, most compression methods convert raw point clouds into octree structures. By recursively dividing space into eight subcells at each level, octrees provide a compact and hierarchical representation of geometry. The maximum level $L$ of the octree controls reconstruction fidelity. With octree, existing codecs can perform progressive, dense-to-sparse predictive coding of occupancy codes, modeling the distribution by exploiting contexts from encoded sibling nodes and ancestral nodes to minimize storage.

Despite the compact and regular structure of octree representations, they face the challenge of HRCS in encoding high-resolution LiDAR point clouds. To address this problem, we propose a **Geometry Re-Densification (GRED)** module and integrate it into the dense-to-sparse progressive coding pipeline. At each HRCS-affected level, GRED downsamples the sparse occupancy codes into denser representations to enhance local context extraction, and then reverts to the original sparse domain for effective prediction and entropy coding. Concretely, for each HRCS-affected level, the module performs the following steps:

1. *Re-Densification.* Downsample the occupancy codes of the last encoded level into a denser octree level, producing an aligned dense feature map with zero-padding for empty nodes.

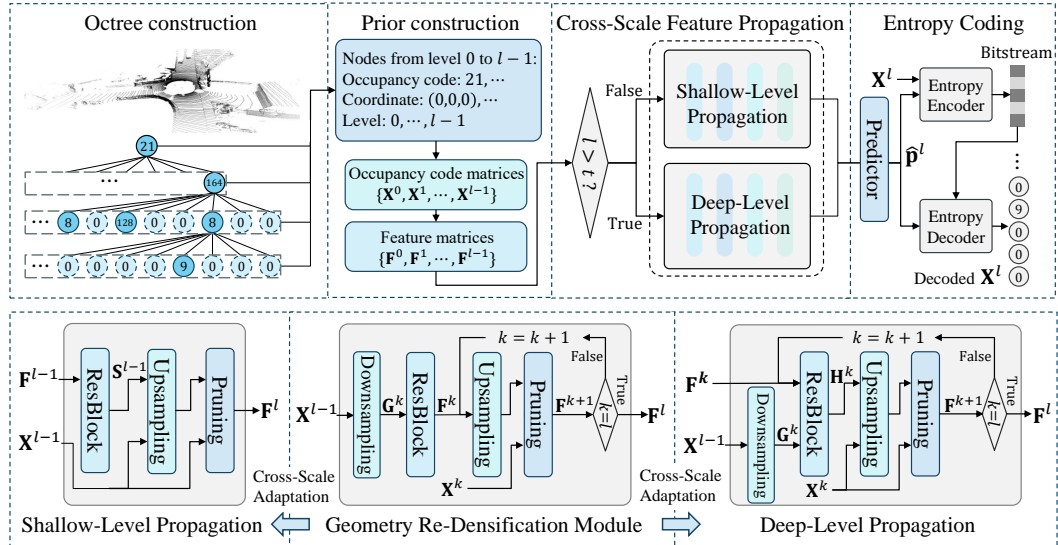

Figure 2: Pipeline of compressing a single octree level in the proposed LiDAR PCC framework. The framework consists of four main stages: octree construction, prior construction, cross-scale feature propagation, and entropy coding. The cross-scale feature propagation module comprises two key components: the shallow-level propagation block and the deep-level propagation block, both adapted from the geometry re-densification module to exploit cross-scale features.

2. *Feature Extraction.* Apply lightweight convolutions to the dense feature map to extract rich local spatial representations.

3. *Re-Sparsification.* Recursively upsample and prune the dense features using the encoded occupancy codes, producing a sparse feature map aligned with the nodes at current level.

4. *Prediction & Coding.* Use a multilayer perceptron (MLP)-based predictor on the sparse feature map to estimate the occupancy distribution over 255 classes, and encode the true occupancy codes using the predicted distribution.

Without loss of generality, we denote octree as $\mathbf{X} = \{\mathbf{X}^1, \mathbf{X}^2, \ldots, \mathbf{X}^L\}$, where $L$ is the maximum level of octree, $\mathbf{X}^l \in \{1, \ldots, 255\}^{N^l}$ represents the occupancy sequence of all nodes at level $l$, and $N^l$ denotes the number of nodes at that level. Lossless compression aims to approximate the true occupancy distribution $P(\mathbf{X})$ with an estimated distribution $Q(\mathbf{X})$ by minimizing the cross-entropy:

$$H(P, Q) = \mathbb{E}_{P(\mathbf{X})} \left[ - \log Q(\mathbf{X}) \right]. \tag{1}$$

Standard octree-based codecs typically estimate the distribution of occupancy codes in a layer-wise autoregressive manner, *i.e.*, previously encoded levels serve as priors for predicting the current one:

$$Q(\mathbf{X}) = \prod_{l=1}^{L} Q\left(\mathbf{X}^l \mid \mathbf{X}^{1:l-1}\right), \tag{2}$$

where each conditional distribution is predicted by an occupancy predictor:

$$Q\left(\mathbf{X}^l \mid \mathbf{X}^{1:l-1}\right) = \text{Predictor}\left(\mathbf{X}^{1:l-1}\right). \tag{3}$$

Different methods design various predictors to estimate this conditional distribution, often leveraging spatial context or learned priors. However, as illustrated in Fig. 1c, this stage is exactly where the HRCS problem emerges, limiting the predictor's ability to make accurate estimations.

Suppose predictions are being made at level $l$. Given the encoded occupancy codes $\{\mathbf{X}^k, \cdots, \mathbf{X}^{l-1}\}$, to obtain denser context features, GRED first downsamples $\mathbf{X}^{l-1}$ into a pre-defined dense octree level $k$:

$$\mathbf{G}^k = \text{Downsampling}\left(\mathbf{X}^{l-1}\right), \tag{4}$$

where the $\text{Downsampling}(\cdot)$ operation embeds the occupancy codes of $l-1$ into feature maps at level $k$ using sparse convolutions. In $\mathbf{G}^k$, each channel corresponds to a specific occupancy state of a node at level $l-1$, thereby enabling *Re-Densification* to enrich the context information with higher density.

To extract contextual features, $\mathbf{G}^k$ is fed to a ResBlock for *Feature Extraction*, thereby obtaining the feature $\mathbf{F}^k$:

$$\mathbf{F}^k = \text{ResBlock}\left(\mathbf{G}^k\right). \tag{5}$$

Although it is possible to directly predict occupancy in this dense space, it would incur prohibitive computational costs due to the vast number of potential sub-nodes. Instead, GRED progressively reverts $\mathbf{F}^k$ to the original sparse space $\mathbf{F}^l$ through multi-step upsampling, thereby achieving the *Re-Sparsification* of the features:

$$\mathbf{F}^{k+1} = \text{Pruning}\left(\text{Upsampling}\left(\mathbf{F}^k\right), \mathbf{X}^k\right), \tag{6}$$

where $\text{Upsampling}(\cdot)$ is a linear transformation followed by a PReLU activation, performing an $8\times$ channel expansion, and $\text{Pruning}(\cdot)$ discards features of unoccupied child nodes. This step is recursively applied until the feature map $\mathbf{F}^l$ is obtained. This feature is then upsampled and fed into an MLP-based predictor to estimate the occupancy distribution:

$$\hat{\mathbf{p}}^l = \text{Predictor}\left(\mathbf{F}^l\right). \tag{7}$$

Then the true occupancy codes $\mathbf{X}^l$ are entropy-encoded using $\hat{\mathbf{p}}^l$, finishing *Prediction & Coding*. Overall, GRED enriches the available context with low computational overhead, while preserving the progressive reconstruction workflow of the decoder. It embodies the principle of *dense feature, sparse prediction*.

Although many 3D tasks employ densification operations (Choe et al., 2022; Deng et al., 2024), such as quantization and downsampling, before processing and analysis, LiDAR point cloud compression presents a unique constraint: the decoder cannot access the full geometry at the beginning of decoding. Therefore, globally pre-densifying all octree levels is infeasible. This insight, combined with the observed nonlinear drop in occupancy density across octree levels, supports the necessity of on-the-fly re-densification within a progressive octree coding pipeline.

## 3.2 Cross-Scale Feature Propagation Module

While the proposed GRED module effectively mitigates HRCS, we delve into the rich inter-scale contextual dependencies across the octree, such as the geometric context from ancestor nodes, to improve occupancy prediction accuracy. Existing octree-based codecs typically extract features and predict occupancy codes independently at each octree level or within local node windows. However, this per-level processing overlooks the strong contextual dependencies across octree scales, leading to redundant feature extraction and limited compression efficiency.

To fully leverage inter-scale context, we propose a unified **Cross-Scale Feature Propagation (XFP) Module** that (i) directly propagates features across octree levels and (ii) generalizes the core idea of the GRED Module into a broader, multi-scale framework. In fact, the GRED Module can be viewed as a special case of XFP, applied only at the deepest levels. XFP shares features from coarser (shallower) levels with finer (deeper) levels, avoiding redundant feature extraction and enhancing contextual awareness. Overall, XFP leverages the octree's hierarchical structure and sparse convolution to efficiently construct a coherent multi-scale feature representation, which facilitates more accurate occupancy prediction and compact encoding.

Suppose we are predicting the occupancy codes at level $l$, meaning that the feature maps $\{\mathbf{F}^1, \cdots, \mathbf{F}^{l-1}\}$ and occupancy codes $\{\mathbf{X}^1, \cdots, \mathbf{X}^{l-1}\}$ are available. The first step in XFP is to determine an appropriate feature propagation strategy. In this work, we define two propagation regimes based on a pre-defined decision-making level $t$:

1. **Shallow levels** ($l \leq t$): feature propagation is conducted without re-densification, as the geometry remains relatively dense.

2. **Deep levels** ($l > t$): feature propagation incorporates contextual information from level $k$ through occupancy-based re-densification.

**Shallow-Level Propagation**. For levels $l \leq t$, the octree is relatively shallow and the contextual information is sufficiently dense, making the HRCS problem less prominent. At these levels, we adopt a simplified version of the GRED module, omitting the *Re-Densification* step. The *Feature Extraction* and *Re-Sparsification* stages are accordingly adapted to balance computational complexity and processing speed. The specific adaptations are as follows:

In the *Feature Extraction* step, since the re-densified feature $\mathbf{G}$ is omitted, the input is directly the encoded feature from level $l-1$, denoted as $\mathbf{F}^{l-1}$. A ResBlock is then applied to extract features, obtain the representation $\mathbf{S}^{l-1}$:

$$\mathbf{S}^{l-1} = \text{ResBlock}(\mathbf{F}^{l-1}). \tag{8}$$

Next, through one step of *Re-Sparsification*, $\mathbf{S}^{l-1}$ is upsampled to level $l$ to produce the re-sparsified feature $\mathbf{F}^l$:

$$\mathbf{F}^l = \text{Pruning}\left(\text{Upsampling}\left(\text{Concat}\left(\mathbf{S}^{l-1}, \mathbf{X}^{l-1}\right)\right), \mathbf{X}^{l-1}\right), \tag{9}$$

where Concat denotes channel-wise concatenation of matrices. The obtained feature $\mathbf{F}^l$ then undergoes the same *Prediction & Coding* as GRED for occupancy estimation and entropy encoding.

**Deep-Level Propagation with Re-Densification**. For $l > t$, the spatial sparsity makes direct propagation less effective. Therefore, we apply the full GRED module at these levels. However, during this process, we aim to incorporate additional inter-scale contextual information to further enrich the extracted features. As a result, the *Feature Extraction* and *Re-Sparsification* components of GRED are adapted accordingly, as detailed below:

GRED utilizes the re-densified feature $\mathbf{G}^k$ for *Feature Extraction*. To fully leverage the information from the previous scale, we concatenate $\mathbf{G}^k$ with the original feature map $\mathbf{F}^k$ and use ResBlock to obtain the fused representation $\mathbf{H}^k$:

$$\mathbf{H}^k = \text{ResBlock}\left(\text{Concat}(\mathbf{F}^k, \mathbf{G}^k)\right). \tag{10}$$

The fused representation $\mathbf{H}^k$ will replace the original input feature $\mathbf{F}^k$ in the *Re-Sparsification*, enabling the original features of different scales can be fused into the current level:

$$\mathbf{F}^{k+1} = \text{Pruning}\left(\text{Upsampling}\left(\text{Concat}(\mathbf{H}^k, \mathbf{X}^k)\right), \mathbf{X}^k\right), \quad k = t, \dots, l-1. \tag{11}$$

By recursively applying the above process, we obtain the feature $\mathbf{F}^l$, which integrates contextual information from the preceding $l - t$ scales. Finally, *Prediction & Coding* is performed at level $l$ based on the feature $\mathbf{F}^l$.

This cross-scale propagation scheme effectively reuses context-rich features from earlier levels and adapts them to finer resolutions through sparse, occupancy-aware operations. By combining the shallow and deep propagation pathways, the proposed XFP module unifies dense and sparse processing into a single framework, enabling efficient and context-aware feature extraction throughout the octree hierarchy.

## 4 EXPERIMENTS

In this section, we present a comprehensive experimental evaluation of our method, including implementation details, comparative results with state-of-the-art approaches, and ablation studies.

### 4.1 SETTINGS

In this section, we detail the experimental setup, including the benchmark datasets, evaluation metrics, and comparative baselines. The implementation details are provided in the appendix.

**Benchmark Datasets**. Experiments are conducted on two different LiDAR datasets: KITTI (Geiger et al., 2012) and Ford (Pandey et al., 2011) dataset. The KITTI dataset consists of 22 stereo sequences collected by a Velodyne LiDAR scanner across diverse continuous scenes, totaling 43,552 frames. Following Fu et al. (2022), we use sequences #00 to #10 for training and #11 to #21 for testing. The Ford dataset comprises three distinct sequences (#01, #02, and #03), each containing 1,500 frames. Consistent with Song et al. (2023), we use sequence #01 for training, and sequences

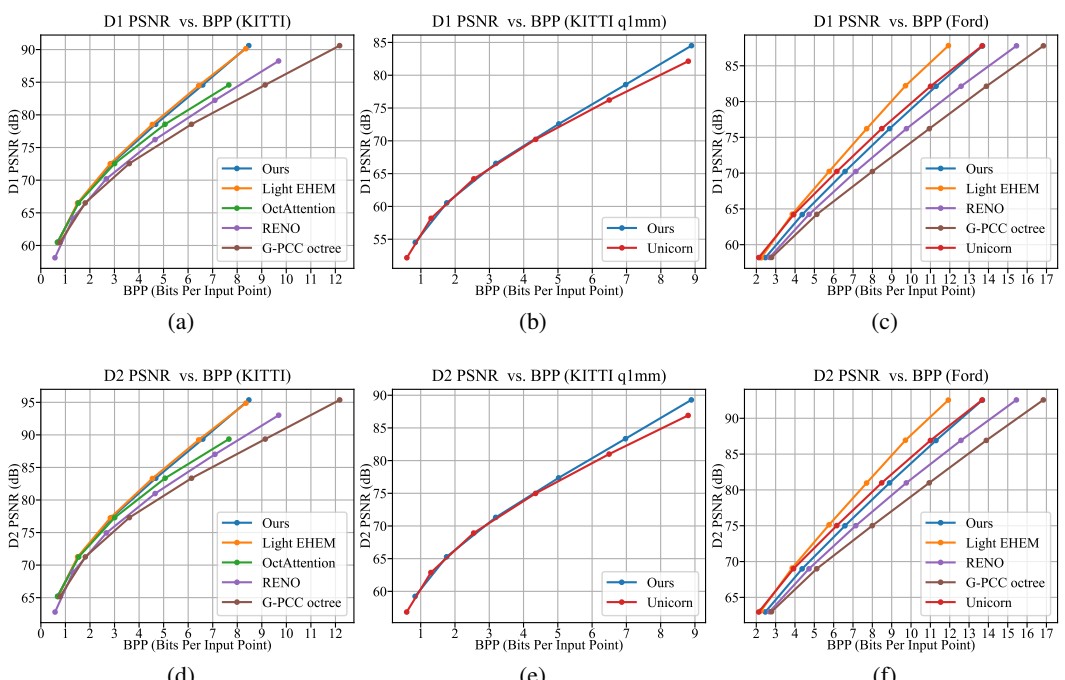

Figure 3: Rate-distortion performance comparison on the KITTI dataset (left two columns) and the Ford dataset (rightmost column).

#02 and #03 for testing. Instead of training on each dataset separately for performance tuning, we adopt joint training across both datasets to enhance generalization ability.

**Evaluation Metrics**. We adopt point-to-point PSNR (D1 PSNR) and point-to-plane PSNR (D2 PSNR) (Tian et al., 2017) for distortion measure. These are standard metrics recommended by MPEG (Schwarz et al., 2019). We employ the Bjøntegaard Delta (BD) metrics (Bjøntegaard, 2001) for evaluating rate-distortion performance, namely Bjøntegaard Delta Peak Signal-to-Noise Ratio (BD-PSNR) and Bjøntegaard Delta Rate (BD-Rate). It is important to note that both BD-Rate and BD-PSNR measure the **relative gains** of a tested model compared to a baseline. A negative BD-Rate or a positive BD-PSNR indicates that the tested model outperforms the baseline.

**Compared Methods**. We compare 5 widely recognized PCC methods. Among them, G-PCC (Schwarz et al., 2019), established by MPEG, serves as the standardized geometry-based benchmark for PCC; OctAttention (Fu et al., 2022) and Light EHEM (Song et al., 2023) represent transformer-driven octree compression method; Unicorn (Wang et al., 2025) is a recent voxel-based PCC method. Finally, RENO (You et al., 2025) introduces an efficient sampling strategy, optimizing the trade-off between compression performance and computational speed. All methods were re-evaluated using their official implementations under standardized experimental conditions, except for EHEM and Unicorn, for which we rely on the originally reported metrics due to the unavailability of their source code.

### 4.2 PERFORMANCE ANALYSIS

This section evaluates the proposed method in terms of rate-distortion performance, computational efficiency, and qualitative visualization, providing a comprehensive assessment of its effectiveness.

**Rate-Distortion Performance**. This section presents the RD performance of the proposed method compared to several existing methods, using two standard evaluation curves: D1 PSNR vs. Bits Per input Point (BPP) and D2 PSNR vs. BPP. A curve closer to the upper-left corner indicates higher reconstruction accuracy at lower bitrates, demonstrating better compression performance. The experimental results are illustrated in Fig. 3. Note that, Unicorn adopts different testing conditions on the KITTI dataset compared to other methods, leading to unaligned metric results. To ensure a fair

Table 1: BD-Rate (%) and BD-PSNR (dB) gains of our model over existing methods.

| Ours vs. Existing Methods | KITTI | | | | Ford | | | |
| --- | --- | --- | --- | --- | --- | --- | --- | --- |
| | BD-Rate (%) | | BD-PSNR (dB) | | BD-Rate (%) | | BD-PSNR (dB) | |
| | D1 | D2 | D1 | D2 | D1 | D2 | D1 | D2 |
| OctAttention (Fu et al., 2022) | -4.815 | -4.869 | 0.644 | 0.650 | - | - | - | - |
| Light EHEM (Song et al., 2023) | 1.405 | 1.398 | -0.142 | -0.143 | 14.444 | 14.920 | -2.314 | -2.375 |
| Unicorn (Wang et al., 2025) | -1.266 | -1.494 | 0.210 | 0.238 | 6.633 | 6.702 | -1.043 | -1.058 |
| RENO (You et al., 2025) | -15.610 | -15.607 | 1.892 | 1.895 | -8.728 | -8.718 | 1.512 | 1.512 |
| G-PCC octree (Schwarz et al., 2019) | -21.949 | -21.972 | 2.536 | 2.545 | -17.039 | -17.033 | 2.997 | 2.998 |

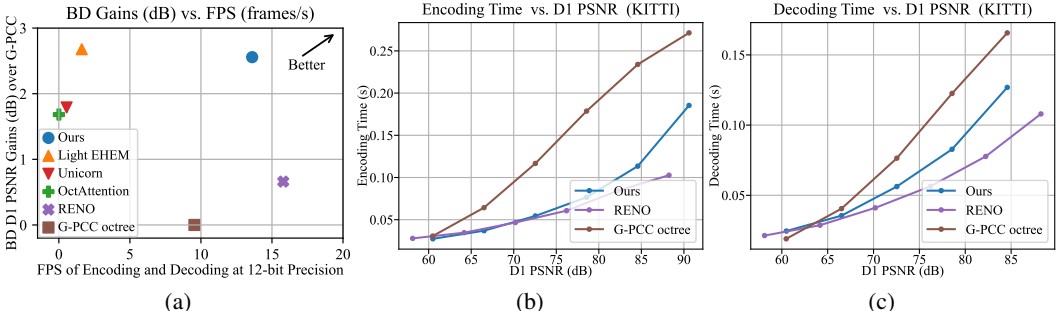

Figure 4: Comparison of encoding time and decoding time on the KITTI dataset.

comparison, we follow the same testing conditions as Unicorn and plot the resulting RD curves in Fig. 3b and Fig. 3e. As observed on the KITTI dataset, our method achieves performance comparable to the transformer-based Light EHEM, and outperforms the recent sparse convolution-based method Unicorn at high bitrates, demonstrating clear advantages in RD performance. Moreover, our method provides substantial computational efficiency improvements over both Light EHEM and Unicorn, as discussed in the Computational Efficiency section. On the Ford dataset, the performance is less favorable, yet our method still outperforms methods with similar coding latency, such as RENO and G-PCC octree. This performance gap may be attributed to the limited number of training samples (only 1,500 frames). Table 1 provides quantitative RD performances of the proposed method over existing methods. On the KITTI dataset, our method achieves an average gain of 2.536 dB (D1 PSNR) and 2.545 dB (D2 PSNR) over G-PCC, as well as 0.210 dB (D1 PSNR) and 0.238 dB (D2 PSNR) improvements over Unicorn. Compared to the efficiency-oriented method RENO, our approach delivers 1.892 dB and 1.895 dB gains in D1 and D2 PSNR, respectively. These results confirm that the proposed effectively exploits redundant information within the octree structure, thereby improving compression performance.

**Computational Efficiency**. To evaluate the real-time performance of the proposed method, we measured the encoding and decoding times of our method and several existing baselines, as summarized in Table 2. The reported times are averaged over 11-bit to 16-bit quantized point clouds of the KITTI test set. The proposed method demonstrates faster runtime than most competing methods. Although slightly slower than the efficiency-oriented method RENO, our method still maintains real-time processing speed while achieving significantly better RD performance. For a more intuitive comparison,

Table 2: Comparison of average encoding time and decoding time across 11-16bits (in seconds).

| Methods | KITTI | | Ford | |
| --- | --- | --- | --- | --- |
| | Enc Time | Dec Time | Enc Time | Dec Time |
| OctAttention (Fu et al., 2022) | 0.229 | 239.250 | - | - |
| Light EHEM (Song et al., 2023) | 0.290 | 0.330 | - | - |
| Unicorn (Wang et al., 2025) | 1.821 | 1.678 | 2.338 | 2.157 |
| RENO (You et al., 2025) | **0.059** | **0.056** | **0.072** | **0.057** |
| G-PCC octree (Schwarz et al., 2019) | 0.149 | 0.103 | 0.150 | 0.107 |
| Ours | 0.082 | 0.089 | 0.103 | 0.112 |

Table 3: BD-Rate (%) and BD-PSNR (dB) gains of the proposed modules on the KITTI dataset.

| Methods | Compared with Baseline | | | | Compared with G-PCC octree | | | |
| | BD-Rate (%) | | BD-PSNR (dB) | | BD-Rate (%) | | BD-PSNR (dB) | |
| | D1 | D2 | D1 | D2 | D1 | D2 | D1 | D2 |
|---|---|---|---|---|---|---|---|---|
| Baseline | 0 | 0 | 0 | 0 | 3.348 | 3.346 | -0.361 | -0.363 |
| + GRED | -8.511 | -8.495 | 0.940 | 0.939 | -5.449 | -5.433 | 0.576 | 0.573 |
| + GRED + XFP | -26.245 | -26.239 | 3.718 | 3.724 | -21.949 | -21.972 | 2.536 | 2.545 |

Fig. 4a plots the BD-PSNR gains against frames per second (FPS) on the 12-bit quantized KITTI test set. Our method reaches 13 FPS for the overall encoding and decoding process, while delivering a BD-PSNR gain of 2.54 dB over G-PCC, surpassing other methods with comparable compression performance. To further evaluate the runtime across different reconstruction qualities, we compare the encoding and decoding times of our method against comparable methods across different quantization precisions. As illustrated in Fig. 4b and Fig. 4c, our method consistently outperforms G-PCC across most settings. At high quantization precision, the runtime becomes slightly longer than that of RENO, likely due to the overhead introduced by the re-densification module. Nevertheless, our method still maintains a competitive speed of approximately 10 FPS, which is sufficient for typical point cloud processing scenarios. Under lower quantization precision, such as 12-bit, our method achieves over 20 FPS for encoding/decoding. This speed is well aligned with the scanning rate of mainstream LiDAR systems, enabling real-time coding of LiDAR point clouds. These results collectively highlight the computational efficiency and practical applicability of the proposed method.

### 4.3 ABLATION STUDIES

To evaluate the individual contributions of the proposed components, we conduct ablation studies on the GRED module and the XFP module. Each component is systematically removed to assess its impact on overall compression performance.

**Ablation of XFP**. To evaluate the effectiveness of the proposed XFP module, we conducted an ablation study by removing the cross-scale features. Quantitative results in Table 3 show that this removal results in a degradation of approximately 2.778 dB (D1 PSNR) and 2.785 dB (D2 PSNR), indicating that the integration of cross-scale information through XFP is critical for improving compression efficiency.

**Ablation of GRED**. To evaluate the effectiveness of the proposed GRED module, we further removed GRED on top of the XFP ablation. In this setting, the dense features extracted from the shallowe level are no longer utilized for predicting the occupancy of deeper levels. As a result, the model is directly exposed to the HRCS problem under high-resolution encoding. Quantitative results in Table 3 show that removing the GRED module leads to a further performance drop of approximately 0.940 dB. This performance gap highlights the positive impact of the GRED module in mitigating the effects of HRCS and enhancing performance.

## 5 CONCLUSION

This paper addresses the challenge of HRCS in LiDAR point cloud compression, which poses a significant obstacle to efficient occupancy prediction at high resolutions. To overcome this issue while achieving real-time processing, we propose a novel compression framework that incorporates the Geometry Re-densification (GRED) module and the Cross-scale Feature Propagation (XFP) module, enabling efficient intra-scale and cross-scale context modeling. Extensive experiments demonstrate that the proposed method achieves superior rate-distortion performance and competitive encoding and decoding speeds, validating its effectiveness in terms of both compression quality and efficiency.

While the proposed framework demonstrates strong performance, it is primarily designed to validate the core ideas of GRED and XFP. To ensure a clear evaluation of these modules, we exclude the cross-scale parameter-sharing strategies used in prior works. In future work, we plan to explore level-aware neural blocks that enable parameter sharing across octree levels, with the goal of enhancing parameter efficiency.

## REPRODUCIBILITY STATEMENT

To facilitate reproducibility, we provide the complete source code of our method in the supplementary materials. Additionally, the appendix includes detailed descriptions of the data preprocessing steps, and the training and testing configurations used in our experiments. These materials together provide the necessary information for reproducing the results presented in the paper.

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

APPENDIX

## A DIFERENCES WITH EXISTING METHODS

To the best of our knowledge, this work is the first to explicitly identify and address the High-Resolution Contextual Sparsity (HRCS) problem in point cloud compression. While it is difficult to rigorously determine whether existing methods have implicitly mitigated HRCS in a generalizable way, we carefully examine several representative approaches to clarify this issue.

**Transformer-based Methods (e.g., EHEM, OctAttention).** These methods represent octree nodes as explicit feature vectors and use transformer architectures to capture long-range dependencies between nodes, thereby enlarging the receptive field. However, it is important to note that such methods essentially process octree nodes in a 1D sequence space, rather than in the native 3D space. While their attention mechanism can implicitly model 3D geometry, it discards the explicit 3D structural information and instead depends heavily on learned embeddings. As a result, these methods typically require large attention windows (e.g., 8192 nodes in Light EHEM) to achieve competitive performance. According to our analysis, this leads to around 10× FLOPs compared to our approach. Thus, while transformer-based methods might sidestep the HRCS problem by modeling 1D node sequence via long-range attention, this comes at the cost of significantly increased computational complexity and latency. Therefore, especially in scenarios where complexity or runtime is a concern, such methods cannot be considered a viable solution to the HRCS problem.

**Sparse Convolution-based Methods (e.g., SparsePCGC, Unicorn).** These methods exploit voxel-level neighborhoods via sparse convolutions in a coarse-to-fine reconstruction pipeline. By design, sparse convolutions skip computation on empty voxels to improve efficiency. However, this sparsity impedes information propagation across voxels when the point cloud is highly sparse at finer scales, making these approaches particularly vulnerable to HRCS. For instance, if an occupied voxel is surrounded by 26 empty neighbors ($3 \times 3 \times 3 - 1$), no amount of stacking $3 \times 3 \times 3$ kernel sparse convolutions can retrieve geometric context for this voxel, making this voxel actually isolated. Although SparsePCGC alleviates this problem to some extent by increasing the depth of the convolution blocks, it suffers from high coding latency. In contrast, our method proposes GRED and XFP modules, which explicitly aim to address HRCS.

In summary, while existing methods may touch on related ideas, none have explicitly recognized HRCS as a core challenge or introduced a targeted solution for it. Our work is, to our knowledge, the first to both formally define HRCS and provide an effective architectural mechanism to overcome it. For clarity, we provide a visualization comparing the context modeling processes of existing methods with our approach in Fig. 5, highlighting the key differences in design.

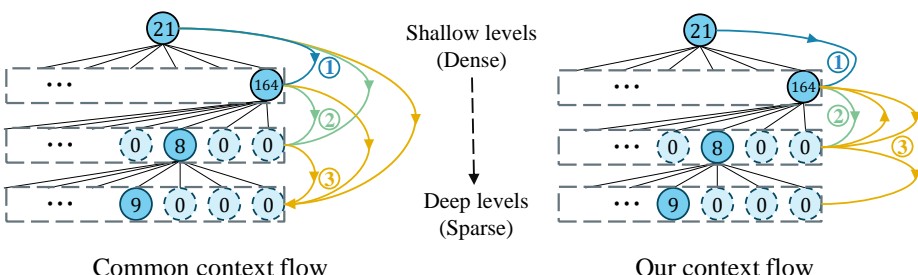

Figure 5: Comparison of context flows in existing octree-based methods (left) and our approach (right). Lines with different colors and numbers indicate different feature extraction and occupancy prediction steps. In common designs, context flow is typically unidirectional, and encoding/decoding at each octree level depends on geometry from multiple preceding levels. By contrast, our approach adopts a bidirectional context flow at deeper octree levels, while at shallow levels, each octree level relies only on the feature and geometry of the immediately preceding level.

## B   DETAILED MODEL STRUCTURE

To provide a clearer understanding of the model structure and workflow of key modules in our proposed network, we present a detailed breakdown of the Upsampling$(\cdot)$, ResBlock$(\cdot)$, Downsampling$(\cdot)$, and Predictor$(\cdot)$ components. These modules are implemented using PyTorch and TorchSparse (Tang et al., 2023), which enable efficient processing of sparse 3D data. The detailed workflow is illustrated in Fig. 6.

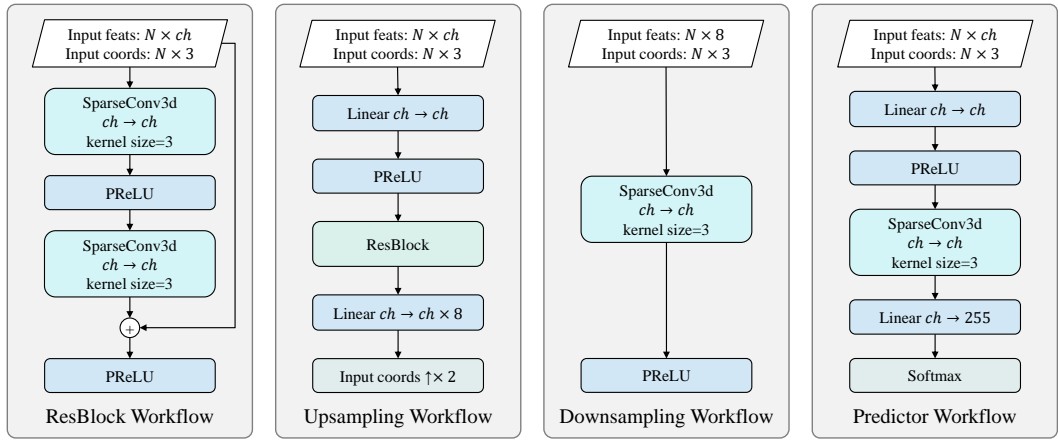

Figure 6:   Illustration of the detailed workflows of the Upsampling$(\cdot)$, ResBlock$(\cdot)$, Downsampling$(\cdot)$, and Predictor$(\cdot)$ modules within the proposed network architecture.

In the current implementation, our primary objective is to validate the core ideas of geometry redensification and cross-scale feature propagation. To ensure an isolated evaluation of the proposed modules, we intentionally omit the cross-scale parameter-sharing mechanisms employed in previous works. Consequently, our model is less parameter-efficient compared to prior methods, as shown in Table 4, since the number of parameters increases linearly with the decrease of the pre-defined minimum octree level ($L-11$ in this paper). For the remaining geometry at this maximum downsampling level, we directly encode the coordinates based on their symbol frequencies. In future work, we plan to improve parameter efficiency by introducing level-aware neural blocks that support parameter sharing across octree levels.

Table 4: Comparison of the number of model parameters.

| Methods | PCGCv2 | OctAttention | EHEM | Ours | RENO |
|---|---|---|---|---|---|
| Number of parameters | 0.77M | 6.99M | 13.01M | 131.89M | 0.28M |

## C   IMPLEMENTATION DETAILS

This section outlines the details of our implementation, including quantization strategies for the KITTI dataset, octree-based operations, training configurations, and evaluation metrics.

### C.1   QUANTIZATION OF KITTI DATASET

The KITTI point clouds are not officially quantized, which has led to two different quantization approaches:

1. The first approach normalizes the point clouds within a bounding box of size $400 \times 400 \times 400$ centered at the origin $(0, 0, 0)$, scales the coordinates by $2^{16}$, and then applies quantization.

2. The second approach, adopted by RENO (You et al., 2025), scales the original floating-point coordinates by 10000, followed by quantization using an additional scale factor $\text{posQ} \in \{8, 16, 32, 64, 128, 256, 512\}$ to generate point clouds of different precision levels.

These two approaches yield point clouds of different fidelity, leading to different PSNR values even under lossless compression settings, where the only distortion arises from the quantization process. Consequently, in Fig. 3a and Fig. 3d, the RD points of RENO do not align with those of other methods, despite all methods being lossless compression methods. While it is technically feasible to unify the quantization strategy, we follow RENO's official setting to report its results, as it better reflects the method's ideal RD performance.

For clarity, we refer to the highest-precision results from both quantization approaches as *16-bit quantization* throughout this paper. In our experiments, the octree level is varied from 11 to 16, enabling control over the RD trade-off by adjusting the spatial resolution.

## C.2 Implementation of Octree Operations

Our model is implemented using PyTorch and TorchSparse. Two components are essential for enabling octree-based operations with sparse convolution: coordinate upsampling/downsampling and occupancy code generation.

**Coordinate Sampling**. To efficiently generate node coordinates for all octree levels, we first sort the input coordinates in Morton order, which exploits the hierarchical spatial locality between Morton codes and octree structures. Starting from the input coordinates, we repeatedly divide them by 2, apply floor rounding, and remove consecutive duplicates. This yields the coordinates of nodes at progressively shallower octree levels. For coordinate upsampling, we reconstruct child node coordinates by adding a pre-defined offset matrix (leveraging matrix broadcasting) to the parent coordinates. We then apply a masking operation to discard coordinates corresponding to unoccupied nodes. Importantly, this process preserves the original Morton order of the coordinate matrix, ensuring perfect alignment between the encoder and decoder without the need for explicit reordering.

**Occupancy Code Generation**. To generate the 0–255 occupancy codes, we apply a fixed-weight sparse convolution with kernel size 2 and stride 2, using an all-one input feature tensor. Each 8-neighbor group (in $2 \times 2 \times 2$) is encoded as an 8-bit occupancy code. For the reverse process, the occupancy code can be efficiently decoded into binary masks using bitwise operations and matrix broadcasting.

## C.3 Training

**Loss Function**. To train the proposed model, we adopt the standard cross-entropy loss, which is widely used in the octree-based PCC. Specifically, the model outputs a probability distribution $\mathbf{Q} \in \mathbb{R}^{N \times 255}$, where each row corresponds to the predicted occupancy probability of one of the $N$ nodes over the 255 possible occupancy codes (from 1 to 255). Let the ground truth occupancy codes be represented by $\mathbf{X} \in \{1, \ldots, 255\}^N$. The loss function is defined as:

$$\mathcal{L}_{CE} = -\frac{1}{N} \sum_{i=1}^{N} \log \mathbf{Q}_{i,\mathbf{X}_i}, \tag{12}$$

where $\mathbf{Q}_{i,\mathbf{X}_i}$ denotes the predicted probability for the ground truth occupancy code $\mathbf{X}_i$ at the $i$-th node. This loss encourages the model to assign higher probabilities to the correct occupancy codes.

**Other Settings**. We adopt the AdamW optimizer (Loshchilov & Hutter, 2019) with a weight decay of 0.0001 and a learning rate of 0.0001. Gradient clipping is applied with a maximum norm threshold of 1.0 to stabilize training. The model is trained for 60 epochs with a batch size of 8. All experiments were conducted on a computer equipped with an AMD EPYC 7R32 CPU and $2\times$ 4090 GPUs. Training takes approximately 4 days on the KITTI dataset and around 6 hours on the Ford dataset.

## D More Quantitative Analysis

In this section, we provide further quantitative analysis to complement the main results.

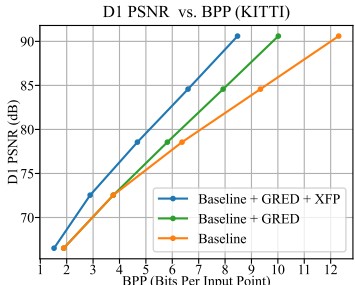 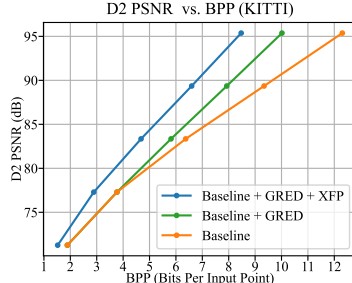

Figure 7: Rate-distortion performance comparison for ablation studies on GRED and XFP modules.

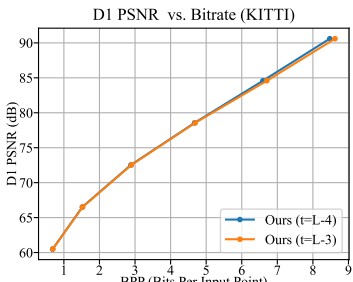 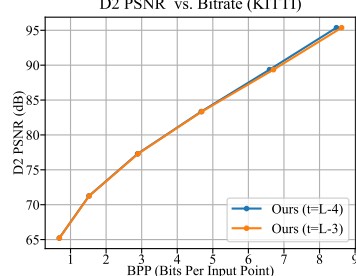

Figure 8: Rate-distortion performance comparison for different choices of $t$ on the KITTI dataset.

### D.1 ABLATION ON GRED AND XFP MODULES

To evaluate the individual contributions of the proposed components, we conduct ablation studies on the GRED and XFP modules. This results in three model variants: "Baseline", "Baseline+GRED", and "Baseline+GRED+XFP". In addition to the RD performance presented in Table 3, we provide the corresponding RD curves in Fig. 7 for visual comparison. The results show that removing XFP leads to a noticeable drop in performance across all bitrates. Removing GRED, on the other hand, only affects performance at higher bitrates (i.e., under high-precision quantization). These observations are consistent with the results in Table 3 and support our analysis regarding the HRCS issue.

### D.2 ABLATION ON THE CHOICE OF $t$

As demonstrated in the Method section, we apply the re-densification module only at levels $l > t$, making $t$ a key hyperparameter. A smaller $t$ causes re-densification to be performed on more levels, enabling more efficient context modeling but at the cost of higher re-densification overhead. Moreover, applying re-densification to shallower levels with dense geometry is often redundant. Therefore, the benefit of reducing $t$ diminishes quickly. In our experiments, unless otherwise stated, we set $t = L - 4$ by default, where $L$ is the deepest octree level. Here, we compare the performance of two settings: $t = L - 4$ and $t = L - 3$. The resulting RD performance is shown in Fig. 8 and Table 5. We can observe slight improvements at high precision when reducing $t$ from $L - 3$ to $L - 4$. However, the average gain across all bitrates is relatively minor. Overall, the setting $t = L - 4$ achieves a 0.76% bitrate reduction on the KITTI dataset.

Table 5: BD gains of the $t = L - 4$ setting over the $t = L - 3$ setting.

| | BD-Rate (%) | | BD-PSNR (dB) | |
| | D1 | D2 | D1 | D2 |
| --- | --- | --- | --- | --- |
| KITTI | -0.760 | -0.759 | 0.093 | 0.093 |

## D.3 COMPUTATIONAL EFFICIENCY

We conducted additional experiments to measure the complexity during the encoding and decoding process, and the results are summarized in Table 6. Note that the FLOPs of sparse convolution depend on the sparsity of the input point clouds, which varies across samples. The reported FLOPs of our model represent the average over the test set of the KITTI dataset. For comparison, we provide the official metrics of Light EHEM in Table 7. Note that the FLOPs reported by Light EHEM are measured per window (with 8192 octree nodes). To ensure a fair comparison, we converted this into an expected average per sample by using the mean octree node count of quantized KITTI point clouds. Based on the summarized results, it is evident that our approach is more efficient in terms of hardware overhead and execution speed compared to the transformer-based method Light EHEM.

Table 6: Detailed complexity metrics of our model on the KITTI dataset.

| Prec. (bits) | Enc Mem (GB) | Dec Mem (GB) | Enc GFLOPs | Dec GFLOPs | Enc Time (s) | Dec Time (s) |
|---|---|---|---|---|---|---|
| 16 | 2.1 | 2.0 | 752.9 | 752.9 | 0.18 | 0.21 |
| 15 | 1.6 | 1.5 | 455.7 | 455.7 | 0.11 | 0.13 |
| 14 | 1.2 | 1.1 | 238.2 | 238.2 | 0.07 | 0.08 |
| 13 | 0.9 | 0.8 | 108.9 | 108.9 | 0.05 | 0.06 |
| 12 | 0.7 | 0.6 | 44.7 | 44.7 | 0.04 | 0.04 |
| 11 | 0.6 | 0.6 | 16.9 | 16.9 | 0.03 | 0.02 |

Table 7: Detailed complexity metrics of Light EHEM on the KITTI dataset.

| Prec. (bits) | Enc/Dec Mem (GB) | GFLOPs per Window | Number of Octree Nodes | Enc/Dec GFLOPs | Enc Time (s) | Dec Time (s) |
|---|---|---|---|---|---|---|
| 16 | 2.6 | 102.9 | 421911.3 | 5299.6 | 1.63 | 1.94 |
| 15 | 2.6 | 102.9 | 302393.4 | 3798.4 | - | - |
| 14 | 2.6 | 102.9 | 191616.0 | 2406.9 | 0.79 | 0.92 |
| 13 | 2.6 | 102.9 | 105002.2 | 1318.9 | - | - |
| 12 | 2.6 | 102.9 | 49486.4 | 621.6 | 0.29 | 0.33 |
| 11 | 2.6 | 102.9 | 20591.8 | 258.7 | - | - |

## D.4 SEQUENCE-WISE PERFORMANCE

Considering the variation in sequence characteristics within the KITTI dataset, we provide sequence-wise RD performance for further analysis and comparison among G-PCC, OctAttention, and our method. The evaluation metrics include D1 PSNR, D2 PSNR, and Chamfer Distance (CD), a widely adopted geometric distortion metric. The results are illustrated in Fig. 9 (sequences #11-#15) and Fig. 10 (sequences #16-#20). These visualizations highlight the performance consistency and robustness of our method across different scenes.

# E QUALITATIVE ANALYSIS

To further evaluate the effectiveness of the proposed method, we visualize compression results at three different bitrates, ranging from low to high. As shown in Fig. 11, the proposed method consistently produces reconstructed point clouds with lower distortion under comparable bit rates. These visual results align well with the findings in the Rate-Distortion Performance section, validating its superiority in compression performance.

# F LLM USAGE STATEMENT

We used a large language model (LLM) solely for language refinement of the manuscript. The LLM did not contribute to the research ideas, experimental design, data analysis, or interpretation of results.

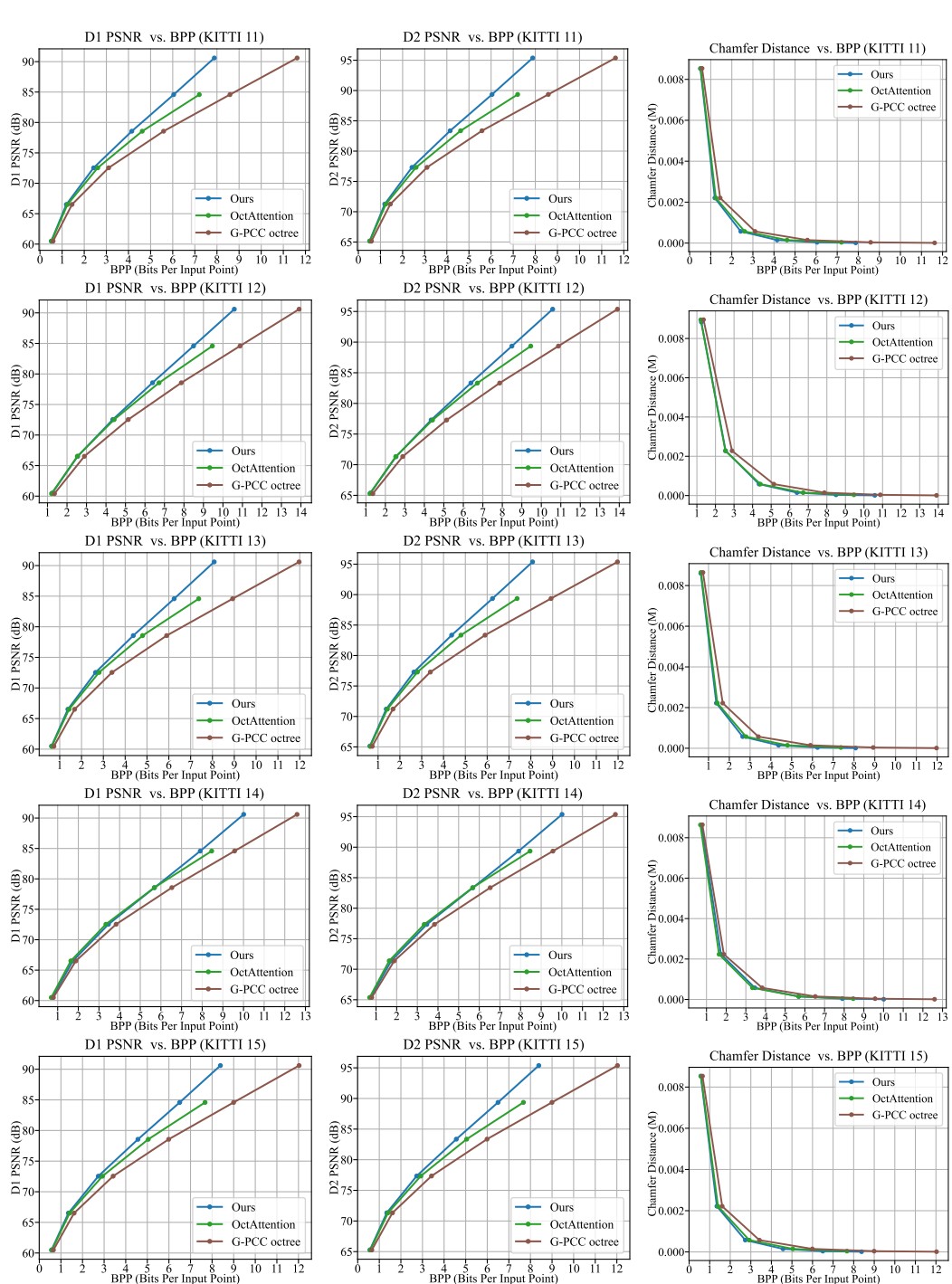

Figure 9: Sequence-wise rate-distortion performance comparison on the KITTI test sequences #11-#15.

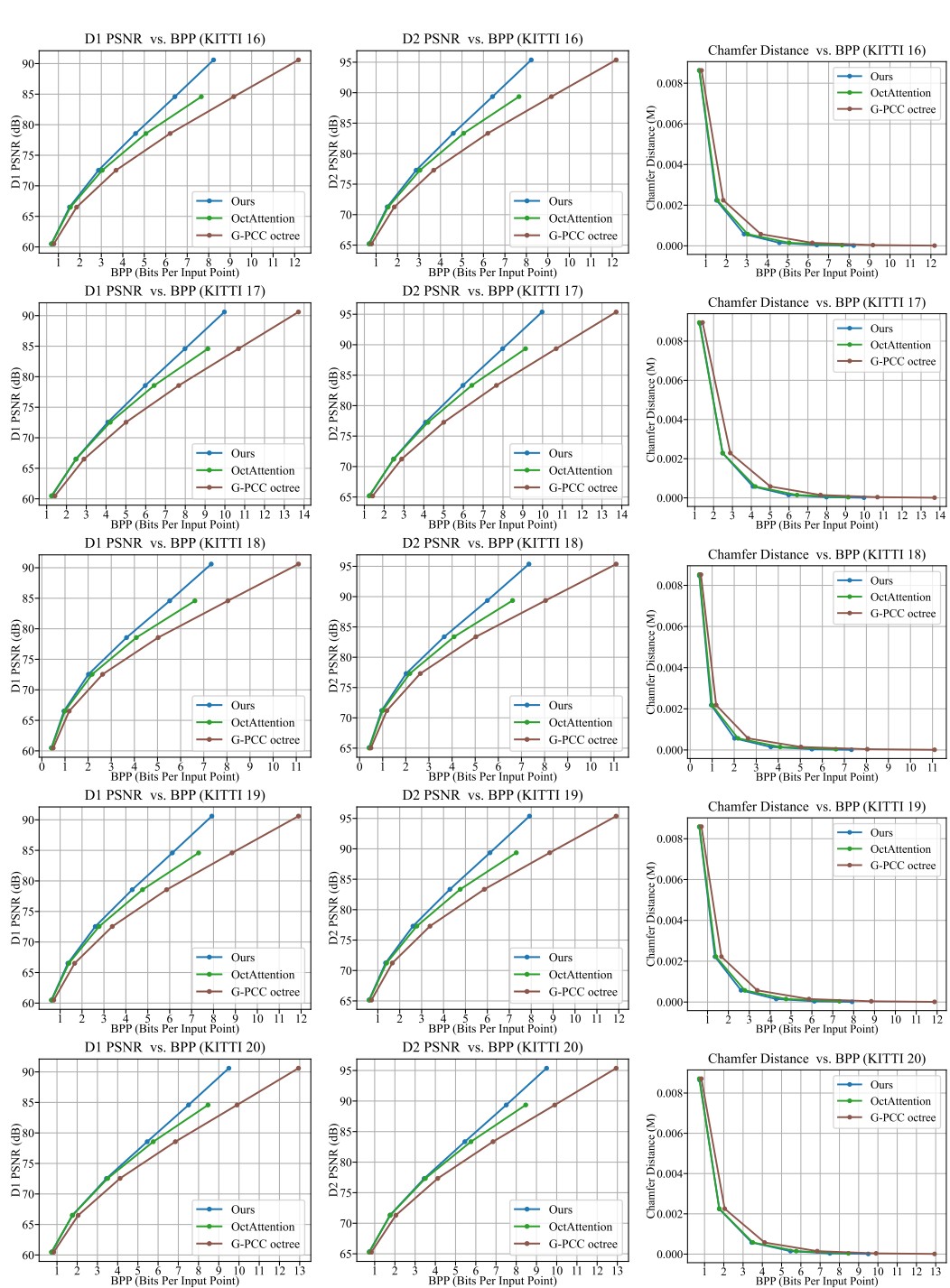

Figure 10: Sequence-wise rate-distortion performance comparison on the KITTI test sequences #16–#20.

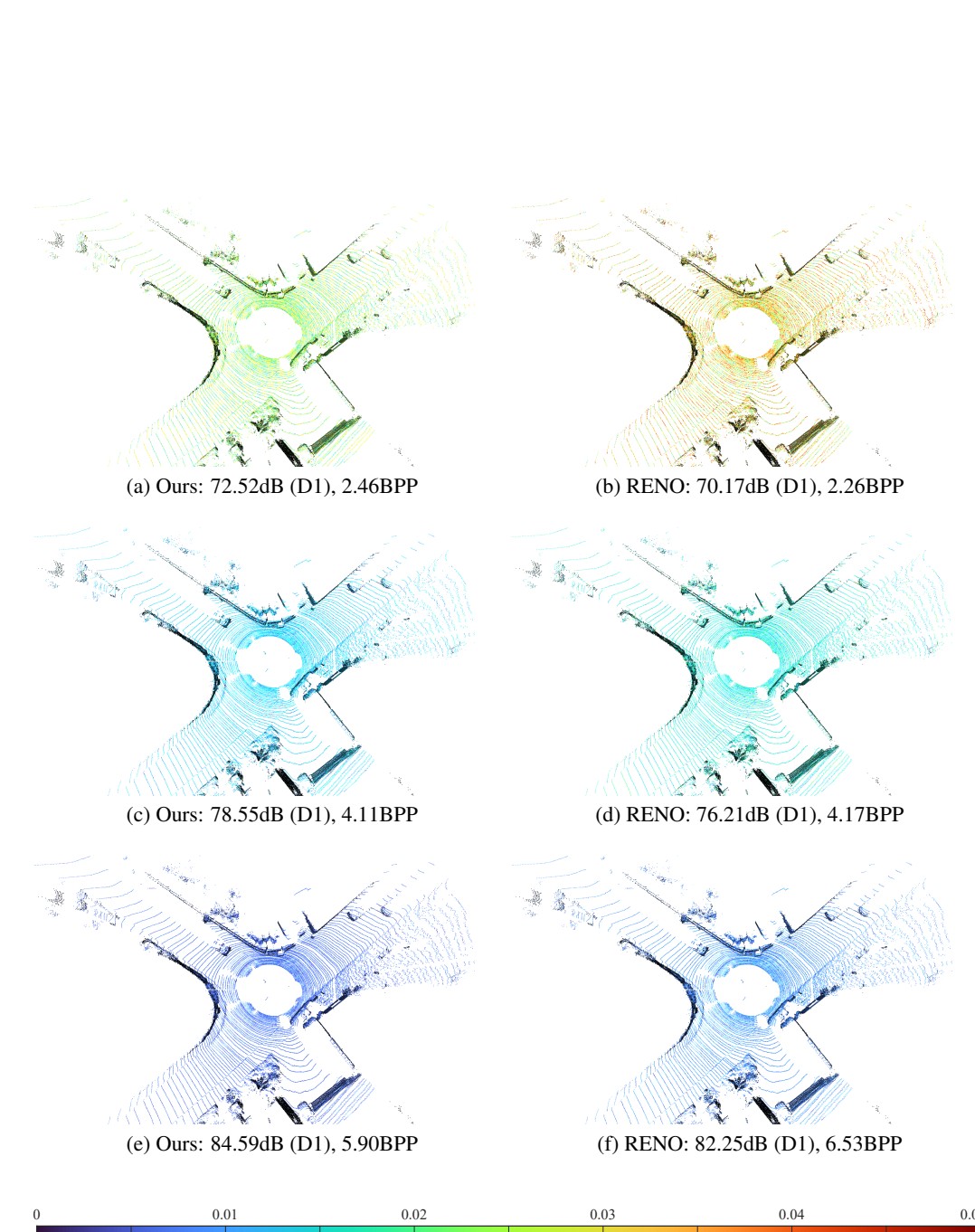

Figure 11: Visualization of reconstruction quality of sample "11_000000.bin" at different quantization precisions.

