# OpenReview forum: "Re-Densification Meets Cross-Scale Propagation: Real-Time Compression of LiDAR Point Clouds"
_ICLR.cc/2026/Conference — ICLR 2026 Conference Withdrawn Submission_

### Official Review · Reviewer_Vzmv · 2025-10-27

**Soundness:** 2
**Presentation:** 2
**Contribution:** 1
**Rating:** 2
**Confidence:** 4

**Summary:**

This paper proposes a point cloud compression method to solve the High-Resolution Contextual Sparsity (HRCS) issue, which means that octree nodes have little neighbors at deep octree levels. To address the HRCS, this paper presents the Geometry Re-Densification (GRED) strategy and the Cross-Scale Feature Propagation (XFP) module. GRED down-samples the high-resolution octree levels to lower resolutions to extract dense features, and then projects these dense features back to higher resolutions. XFP combines dense features from lower resolutions and sparse features at deeper octree levels. The proposed network yields a compression performance comparable to Light-EHEM, while its processing speed is lower than RENO.

**Strengths:**

1. The proposed method is clear and easy to follow.
2. The paper is well organized.

**Weaknesses:**

1. The core contribution of this paper is to solve the HRCS issue in octree-based point cloud compression. However, this issue has been largely addressed by Transformer-based methods (e.g., EHEM and OctAttention). These approaches use a fix-length context window, which ensures the prediction still utilizes plenty of contextual nodes even at high resolutions. Furthermore, existing enhancement-layer-based methods (e.g., GRASP-Net [1]) share a similar idea to re-densification. These methods collect contextual features from the dense base layer, which also alleviates HRCS. These existing solutions weaken the contribution of the work.

2. The proposed method lacks novel modifications upon existing works. GRED is a simple combination of down-sampling and up-sampling blocks, and XFP just additionally concatenates the re-densified features $G_k$ in the propagation pathway. These modifications are not solid enough to constitute strong contributions.

3. Performance improvements are not significant enough. The rate-distortion performance of the proposed method is inferior to the state-of-the-art method EHEM. Although the paper claims an improved processing speed, the runtime of the proposed method is more than 2 times slower than RENO at high resolutions. Therefore, the proposed method just reaches a performance-complexity trade-off between EHEM and RENO, rather than greatly pushing forward the Pareto frontier.

4. The experiment is conducted on only the LiDAR datasets, while the performance on dense point cloud dataset (e.g., 8iVFB and MVUB) has not been evaluated. Complementing experiments on these datasets may enhance the completeness of the work.

5. The authors claim that “Existing octree-based codecs typically extract features and predict occupancy codes independently at each octree level or within local node windows” (line 251), which is overstated. It is common to draw contextual features from ancestor nodes in octree-based methods such as (EHEM and OctSqueeze). Therefore, the contribution of XFP is not as significant as authors’ claim.

6. Although the authors mention that Transformer-based methods suffer from large computational overhead, there have been many solutions (e.g., LinearAttention [2], FlashAttention [3] and KV Cache) to improve Transformer’s efficiency. From this perspective, it seems more effective to directly deploy an accelerated Transformer, rather than proposing another model to speed up coding speed while compromising the rate-distortion performance.

**References**

[1] GRASP-Net: Geometric residual analysis and synthesis for point cloud compression. Proceedings of the 1st International Workshop on Advances in Point Cloud Compression, Processing and Analysis. 2022.

[2] Transformers are RNNs: Fast Autoregressive Transformers with Linear Attention. ICML 2020.

[3] FlashAttention: Fast and Memory-Efficient Exact Attention with IO-Awareness. NeurIPS 2022.

**Questions:**

Since this work does not introduce new ideas on acceleration, does the proposed method rely on the acceleration techniques in RENO to achieve the real-time processing speed? If not, why does the processing speed of the proposed method being faster than Unicorn?

---

### Official Review · Reviewer_Uukh · 2025-10-29

**Soundness:** 3
**Presentation:** 3
**Contribution:** 3
**Rating:** 6
**Confidence:** 3

**Summary:**

The paper targets high-resolution lossless geometry compression of LiDAR point clouds using an octree representation. The authors identify High-Resolution Contextual Sparsity (HRCS)—the observation that, as octree depth increases, local neighborhoods become so sparse that context modeling for occupancy prediction becomes ineffective. To mitigate HRCS, they propose:

1. Geometry Re-Densification (GRED): downsample sparse geometry to a denser (shallower) level to extract local features, then re-sparsify them back to the original level before prediction. The predictor is an MLP over 255 occupancy classes.

2. Cross-Scale Feature Propagation (XFP): a scheme that shares features across octree levels; shallow levels propagate without re-densification, while deep levels incorporate GRED-based densification.

On KITTI and Ford, the method reportedly achieves RD gains over G‑PCC and efficiency‑oriented RENO, with encoding/decoding speeds that the paper better than 10 fps.

**Strengths:**

1. The idea of GRED/XFP are intuitive, and the appendix provides concrete module diagrams, aiding reproducibility.
2. Competitive results with practical speed: On KITTI the method achieves strong BD gains over classical G‑PCC and efficiency‑oriented RENO while keeping per‑frame latency in the tens of milliseconds range.

**Weaknesses:**

1.  Choice of k is not ablated in GRED. the method hinges on how far you densify but sensitivity analysis is missing. It would be good to add sensitivity analysis on k.
2.  How would GRED performs in the dynamic point cloud setting? It would be good to include temporal experiments.

**Questions:**

1. Were other baselines trained with the same joint KITTI+Ford regime?

---

### Official Review · Reviewer_XAMJ · 2025-11-01

**Soundness:** 3
**Presentation:** 3
**Contribution:** 3
**Rating:** 6
**Confidence:** 3

**Summary:**

The primary focus of this paper addresses the ineffectiveness of sparse convolutions when applied to large-scale, sparse LiDAR point clouds. The core contribution is a multi-scale downsampling module designed to efficiently densify these sparse point clouds, enabling more effective feature extraction. Experimentally, the method demonstrates strong performance on the KITTI dataset, though its results on the Ford dataset are more moderate.

**Strengths:**

This work introduces a fast downsampling mechanism that enables real-time compression performance. Furthermore, it achieves highly competitive rate-distortion performance, with results approaching those of Light EHEM on the KITTI dataset and second only to Unicorn on the Ford dataset.

Song, R., Fu, C., Liu, S., & Li, G. (2023). Efficient hierarchical entropy model for learned point cloud compression. In Proceedings of the IEEE/CVF Conference on Computer Vision and Pattern Recognition (pp. 14368-14377).
Wang, J., Xue, R., Li, J., Ding, D., Lin, Y., & Ma, Z. (2024). A versatile point cloud compressor using universal multiscale conditional coding–Part I: Geometry. IEEE transactions on pattern analysis and machine intelligence.

**Weaknesses:**

1：The primary motivation of this work is to address the problem of insufficient receptive fields for sparse convolutions in the context of sparse LiDAR point clouds. However, this is not the first paper to identify this issue. For instance, prior works like Grasp-Net and Unicorn have also observed this phenomenon and have implemented corresponding measures to address it.
2：While the authors' proposed multi-scale downsampling module is efficient, its performance is notably inconsistent across different datasets. This leads me to be concerned that such a design may be highly sensitive to varying types of sparse point cloud data and may not possess good generalization capabilities.

Pang, Jiahao, Muhammad Asad Lodhi, and Dong Tian. "GRASP-Net: Geometric residual analysis and synthesis for point cloud compression." Proceedings of the 1st International Workshop on Advances in Point Cloud Compression, Processing and Analysis. 2022.
Wang, J., Xue, R., Li, J., Ding, D., Lin, Y., & Ma, Z. (2024). A versatile point cloud compressor using universal multiscale conditional coding–Part I: Geometry. IEEE transactions on pattern analysis and machine intelligence.

**Questions:**

NA

---

### Note · Authors · 2025-11-14

I have read and agree with the venue's withdrawal policy on behalf of myself and my co-authors.